# PHRASE-BASED ATTENTIONS

## ABSTRACT

Most state-of-the-art neural machine translation systems, despite being different in architectural skeletons (*e.g.,* recurrence, convolutional), share an indispensable feature: the Attention. However, most existing attention methods are token-based and ignore the importance of phrasal alignments, the key ingredient for the success of phrase-based statistical machine translation. In this paper, we propose novel phrase-based attention methods to model n-grams of tokens as attention entities. We incorporate our phrase-based attentions into the recently proposed Transformer network, and demonstrate that our approach yields improvements of up to 1.3 and 0.5 BLEU points in English-to-German and German-to-English translation tasks, and 1.75 and 1.35 BLEU points in English-to-Russian and Russian-to-English translation tasks on WMT'14 test set using WMT'16 training data.

## 1 INTRODUCTION

**Neural Machine Translation (NMT)** has established breakthroughs in many different translation tasks, and has quickly become the standard approach to machine translation. NMT offers a simple encoder-decoder architecture that is trained end-to-end. Most NMT models (except a few like (Kaiser & Bengio, 2016) and (Huang et al., 2018)) possess attention mechanisms to perform alignments of the target tokens to the source tokens. The attention module plays a role analogous to the word alignment model in Statistical Machine Translation or SMT (Koehn, 2010). In fact, the Transformer network introduced recently by Vaswani et al. (2017) achieves state-of-the-art performance in both speed and BLEU scores (Papineni et al., 2002) by using only attention modules.

On the other hand, phrasal interpretation is an important aspect for many language processing tasks, and forms the basis of Phrase-Based Machine Translation (Koehn, 2010). Phrasal alignments (Koehn et al., 2003) can model one-to-one, one-to-many, many-to-one, and many-to-many relations between source and target tokens, and use local context for translation. They are also robust to non-compositional phrases. Despite the advantages, the concept of **phrasal attentions** has largely been neglected in NMT, as most NMT models generate translations token-by-token autoregressively, and use the token-based attention method which is order invariant. Therefore, the intuition of phrase-based translation is vague in existing NMT systems that solely depend on the underlying neural architectures (recurrent, convolutional, or self-attention) to incorporate contextual information. However, the information aggregation strategies employed by the underlying neural architectures provide context-relevant clues only to represent the current token, and do not explicitly model phrasal alignments. We argue that having an explicit inductive bias for phrases and phrasal alignments is necessary for NMT to exploit the strong correlation between source and target phrases.

In this paper, we propose phrase-based attention methods for phrase-level alignments in NMT. Specifically, we propose two novel phrase-based attentions, namely **CONVKV** and **QUERYK**, designed to assign attention scores directly to phrases in the source and compute phrase-level attention vector for the target. We also introduce three new attention structures, which apply these methods to conduct phrasal alignments. Our **homogeneous** and **heterogeneous** attention structures perform *token-to-token* and *token-to-phrase* mappings, while the **interleaved heterogeneous** attention structure models all *token-to-token, token-to-phrase, phrase-to-token*, and *phrase-to-phrase* alignments.

To show the effectiveness of our approach, we apply our phrase-based attention methods to all multi-head attention layers of the Transformer. Our experiments on WMT'14 translation tasks show improvements of up to **1.3** and **0.5** BLEU points for English-to-German and German-to-English

respectively, and up to **1.75** and **1.35** BLEU points for English-to-Russian and Russian-to-English respectively, compared to the baseline Transformer network trained in identical settings.

## 2 BACKGROUND

Most NMT models adopt an encoder-decoder framework, where the encoder network first transforms an input sequence of symbols $\boldsymbol{x} = (x_1, x_2, \ldots, x_n)$ to a sequence of continuous representations $\boldsymbol{Z} = (\boldsymbol{z}_1, \boldsymbol{z}_2, \ldots, \boldsymbol{z}_n)$, from which the decoder generates a target sequence of symbols $\boldsymbol{y} = (y_1, y_2, \ldots, y_m)$ autoregressively, one element at a time. Recurrent seq2seq models with diverse structures and complexity (Sutskever et al., 2014; Bahdanau et al., 2015; Luong et al., 2015; Wu et al., 2016) are the first to yield state-of-the-art results. Convolutional seq2seq models (Kalchbrenner et al., 2016; Gehring et al., 2017; Kaiser et al., 2018) alleviate the drawback of sequential computation of recurrent models and leverage parallel computation to reduce training time.

The recently proposed **Transformer** network (Vaswani et al., 2017) structures the encoder and the decoder entirely with stacked self-attentions and cross-attentions (only in the decoder). In particular, it uses a multi-headed, scaled multiplicative attention defined as follows:

$$\text{Attention}(\boldsymbol{Q}, \boldsymbol{K}, \boldsymbol{V}, \boldsymbol{W}_q, \boldsymbol{W}_k, \boldsymbol{W}_v) = \mathcal{S}(\frac{(\boldsymbol{Q}\boldsymbol{W}_q)(\boldsymbol{K}\boldsymbol{W}_k)^T}{\sqrt{d_k}})(\boldsymbol{V}\boldsymbol{W}_v) \tag{1}$$

$$\text{Head}^i = \text{Attention}(\boldsymbol{Q}, \boldsymbol{K}, \boldsymbol{V}, \boldsymbol{W}_q^i, \boldsymbol{W}_k^i, \boldsymbol{W}_v^i) \text{ for } i = 1 \ldots h \tag{2}$$

$$\text{AttentionOutput}(\boldsymbol{Q}, \boldsymbol{K}, \boldsymbol{V}, \boldsymbol{W}) = \text{concat}(\text{Head}^1, \text{Head}^2, \ldots, \text{Head}^h)\boldsymbol{W} \tag{3}$$

where $\mathcal{S}$ is the *softmax* function, $\boldsymbol{Q}$, $\boldsymbol{K}$, $\boldsymbol{V}$ are the matrices with query, key, and value vectors, respectively, $d_k$ is the dimension of the query/key vectors; $\boldsymbol{W}_q^i$, $\boldsymbol{W}_k^i$, $\boldsymbol{W}_v^i$ are the head-specific weights for query, key, and value vectors, respectively; and $\boldsymbol{W}$ is the weight matrix that combines the outputs of the heads. The attentions in the encoder and decoder are based on **self-attention**, where all of $\boldsymbol{Q}$, $\boldsymbol{K}$ and $\boldsymbol{V}$ come from the output of the previous layer. The decoder also has **cross-attention**, where $\boldsymbol{Q}$ comes from the previous decoder layer, and the $\boldsymbol{K}$-$\boldsymbol{V}$ pairs come from the encoder. We refer readers to (Vaswani et al., 2017) for further details of the network design.

One crucial issue with the attention mechanisms employed in the Transformer network as well as other NMT architectures (Luong et al., 2015; Gehring et al., 2017) is that they are order invariant locally and globally. That is, changing the order of the vectors in Q, K and V does not change the resulted attention weights and vectors. If this problem is not tackled properly, the model may not learn the sequential characteristics of the data. RNN-based models (Bahdanau et al., 2015; Luong et al., 2015) tackle this issue with a recurrent encoder and decoder, CNN-based models like (Gehring et al., 2017) use position embeddings, while the Transformer uses positional encoding. Another limitation is that these attention methods attend to tokens, and play a role analogous to word alignment models in traditional SMT. It is, however, well admitted in SMT that phrases are better than words as translation units (Koehn, 2010). Without explicit attention to phrases, a particular attention function has to depend entirely on the token-level *softmax* scores of a phrase for phrasal alignment, which is not robust and reliable, thus making it more difficult for the model to learn the required mappings. For example, the attention heatmaps of the Transformer (Vaswani et al., 2017) show concentration of the scores on individual tokens even if it uses multiple heads concurrently in multiple layers. Our main hypothesis is that in order to exploit the strong correlation between source and target phrases, the NMT models should have explicit inductive biases towards phrases.

There exists some research on phrase-based decoding in NMT framework. For example, Huang et al. (2018) proposed a phrase-based decoding approach based on a soft reordering layer and a Sleep-WAke Network (SWAN), a segmentation-based sequence model proposed by Wang et al. (2017a). Their decoder uses a recurrent architecture without any attention on the source. Tang et al. (2016) and Wang et al. (2017b) used an external phrase memory to decode phrases for a Chinese-to-English translation task. In addition, hybrid search and PBMT were introduced to perform phrasal translation in (Dahlmann et al., 2017). Nevertheless, to the best of our knowledge, our work is the first to embed phrases into attention modules, which thus propagate the information throughout the entire end-to-end Transformer network, including the encoder, decoder, and the cross-attention.

## 3 MULTI-HEAD PHRASE-BASED ATTENTION

In this section, we present our proposed methods to compute attention weights and vectors based on n-grams of queries, keys, and values. We compare and discuss the pros and cons of these methods. For simplicity, we describe them in the context of the Transformer network; however, it is straightforward to apply them to other architectures such as RNN-based or CNN-based seq2seq models.

### 3.1 PHRASE-BASED ATTENTION METHODS

In this subsection, we present two novel methods to achieve phrasal attention. In Subsection 3.2, we present our methods for combining different types of n-gram attentions. The key element in our methods is a temporal (or one-dimensional) convolutional operation that is applied to a sequence of vectors representing tokens. Formally, we can define the **convolutional** operator applied to each token $x_t$ with corresponding vector representation $\mathbf{x}_t \in \mathbb{R}^{d_1}$ as:

$$o_t = \mathbf{w} \oplus_{k=0}^{n} \mathbf{x}_{t \pm k} \tag{4}$$

where $\oplus$ denotes vector concatenation, $\mathbf{w} \in \mathbb{R}^{n \times d_1}$ is the weight vector (*a.k.a.* kernel), and $n$ is the window size. We repeat this process with $d_2$ different weight vectors to get a $d_2$-dimensional latent representation for each token $x_t$. We will use the notation $\text{Conv}_n(\boldsymbol{X}, \boldsymbol{W})$ to denote the convolution operation over an input sequence $\boldsymbol{X}$ with window size $n$ and kernel weights $\boldsymbol{W} \in \mathbb{R}^{n \times d_1 \times d_2}$.

#### 3.1.1 KEY-VALUE CONVOLUTION

The intuition behind **key-value convolution** technique is to use trainable kernel parameters $\boldsymbol{W}_k$ and $\boldsymbol{W}_v$ to compute the latent representation of n-gram sequences using convolution operation over key and value vectors. The attention function with key-value convolution is defined as:

$$\text{CONVKV}(\boldsymbol{Q}, \boldsymbol{K}, \boldsymbol{V}) = \mathcal{S}\left(\frac{(\boldsymbol{Q}\boldsymbol{W}_q)\text{Conv}_n(\boldsymbol{K}, \boldsymbol{W}_k)^T}{\sqrt{d_k}}\right) \text{Conv}_n(\boldsymbol{V}, \boldsymbol{W}_v) \tag{5}$$

where $\mathcal{S}$ is the *softmax* function, $\boldsymbol{W}_q \in \mathbb{R}^{d_q \times d_k}, \boldsymbol{W}_k \in \mathbb{R}^{n \times d_k \times d_k}, \boldsymbol{W}_v \in \mathbb{R}^{n \times d_v \times d_v}$ are the respective kernel weights for $\boldsymbol{Q}$, $\boldsymbol{K}$ and $\boldsymbol{V}$. Throughout this paper, we will use $\mathcal{S}$ to denote the *softmax* function. Note that in this convolution, the key and value sequences are left zero-padded so that the sequence length is preserved after the convolution (*i.e.,* one latent representation per token).

The CONVKV method can be interpreted as *indirect* query-key attention, in contrast to the *direct* query-key approach to be described next. This means that the queries do not interact directly with the keys to learn the attention weights; instead the model relies on the kernel weights ($\boldsymbol{W}_k$) to learn n-gram patterns.

#### 3.1.2 QUERY-AS-KERNEL CONVOLUTION

In order to allow the queries to *directly* and *dynamically* influence the word order of phrasal keys and values, we introduce **Query-as-Kernel** attention method. In this approach, when computing the attention weights, we use the query as kernel parameters in the convolution applied to the series of keys. The attention output in this approach is given by:

$$\text{QUERYK}(\boldsymbol{Q}, \boldsymbol{K}, \boldsymbol{V}) = \mathcal{S}\left(\frac{\text{Conv}_n(\boldsymbol{K}\boldsymbol{W}_k, \boldsymbol{Q}\boldsymbol{W}_q)}{\sqrt{d_k * n}}\right) \text{Conv}_n(\boldsymbol{V}, \boldsymbol{W}_v) \tag{6}$$

where $\boldsymbol{W}_q \in \mathbb{R}^{n \times d_q \times d_k}, \boldsymbol{W}_k \in \mathbb{R}^{d_k \times d_k}, \boldsymbol{W}_v \in \mathbb{R}^{n \times d_v \times d_v}$ are trainable weights. Notice that we include the window size $n$ (phrase length) in the scaling factor to counteract the fact that there are $n$ times more multiplicative operations in the convolution than the traditional matrix multiplication.

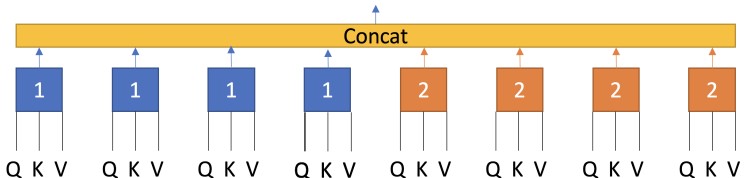

Figure 1: Homogeneous multi-head attention, where each attention head features one n-gram type. In this example, there are eight heads, which are distributed equally between unigrams and bigrams.

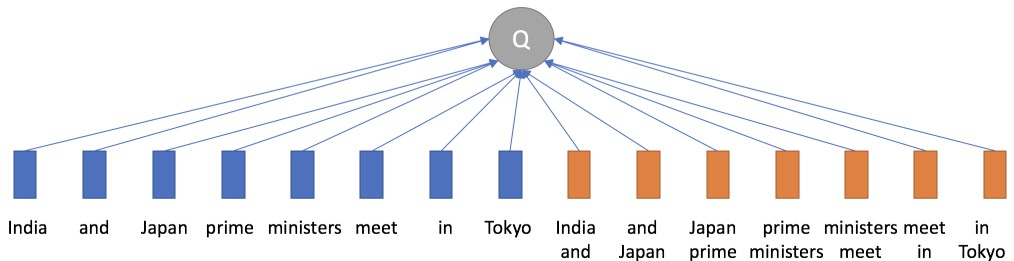

Figure 2: Heterogeneous n-gram attention for each attention head. Attention weights and vectors are computed from all n-gram types simultaneously.

## 3.2 MULTI-HEADED PHRASAL ATTENTION

Having presented the two phrase-based attention methods in the previous subsection, we now introduce our extensions to the multi-headed attention framework of the Transformer to enable it to pay attention not only to tokens but also to phrases across many sub-spaces and locations.

### 3.2.1 HOMOGENEOUS N-GRAM ATTENTION

In **homogeneous n-gram attention**, we distribute the attention heads to different n-gram types with each head attending to one particular n-gram type (n = $1, 2, \ldots, N$). For instance, Figure 1 shows a homogeneous structure, where the first four heads attend to unigrams, and the last four attend to bigrams. A head can apply one of the phrasal attention methods described in Subsection 3.1. The selection of which n-gram to assign to how many heads is considered as hyperparameters to the model. Since all heads must have consistent sequence length, phrasal attention heads in the homogeneous setting require left-padding of keys and values before convolution.

Since each head attends to a subspace resulting from one type of n-gram, homogeneous attention learns the mappings in a distributed way. However, the homogeneity restriction may limit the model to learn interactions between different n-gram types since the gradients for different n-gram types flow in parallel paths with no explicit interactions. Furthermore, the homogeneous heads force the model to assign each query with attentions on all n-gram types (*e.g.,* unigrams and bigrams) even when it does not need to do so, thus possibly inducing more noise into the model.

### 3.2.2 HETEROGENEOUS N-GRAM ATTENTION

The **heterogeneous n-gram attention** relaxes the constraint of the homogeneous approach. Instead of limiting each head's attention to a particular type of n-gram, it allows the query to freely attend to all types of n-grams simultaneously. To achieve this, we first compute the attention logit for each n-gram type separately (*i.e.,* for n = $1, 2, \ldots, N$), then we concatenate all the logits before passing them through the *softmax* layer to compute the attention weights over all n-gram types. Similarly, the value vectors for the n-gram types are concatenated to produce the overall attention output. Figure 2 demonstrates the heterogeneous attention process for unigrams and bigrams.

For CONVKV technique in Equation 5, the attention output is given by:

$$\mathcal{S}(\frac{(\boldsymbol{Q}\boldsymbol{W}_q)[(\boldsymbol{K}\boldsymbol{W}_{k,1})^T; \mathrm{Conv}_2(\boldsymbol{K}, \boldsymbol{W}_{k,2})^T; ...]}{\sqrt{d_k}})[(\boldsymbol{V}\boldsymbol{W}_{v,1}); \mathrm{Conv}_2(\boldsymbol{V}, \boldsymbol{W}_{v,2}); ...] \qquad (7)$$

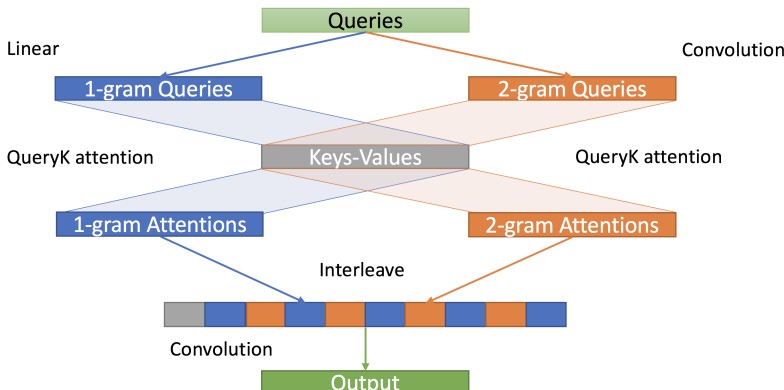

Figure 3: Interleaved phrase-to-phrase heterogeneous attention. The queries are first transformed into unigram and bigram representations, which in turn then attend independently on key-value pairs to produce unigram and bigram attention vectors. The attention vectors are then interleaved before passing through another convolutional layer.

For QUERYK technique (Equation 6), the attention output is given as follows:

$$\mathcal{S}([\frac{(\boldsymbol{QW}_{q,1})(\boldsymbol{KW}_{k,1})^T}{\sqrt{d}}; \frac{\text{Conv}_2(\boldsymbol{KW}_{k,2}, \boldsymbol{QW}_{q,2})}{\sqrt{d*n_2}}; ...])[(\boldsymbol{VW}_{v,1}); \text{Conv}_2(\boldsymbol{V}, \boldsymbol{W}_{v,2}); ...] \tag{8}$$

Note that in heterogeneous attention, we do not need to pad the input sequences before the convolution operation to ensure identical sequence length. Also, the key/value sequences that are shorter than the window size do not have any valid phrasal component to be attended.

### 3.3 INTERLEAVED PHRASES TO PHRASE HETEROGENEOUS ATTENTION

All the methods presented above perform attention mappings from token-based queries to phrase-based key-value pairs. In other words, they model token-to-token and token-to-phrase structures. These types of attentions are beneficial when there exists a translation of a phrase in the source language (keys and values) to a single token in the target language (query). However, these methods are not explicitly designed to work in the reverse direction when a phrase or a token in the source language should be translated to a phrase in the target language. In this section, we present a novel approach to heterogeneous phrasal attention that allows phrasal queries to attend to tokens and phrases of keys and values (*i.e.,* phrase-to-token and phrase-to-phrase mappings).

We accomplish this with the QUERYK and CONVKV methods as follows. We first apply convolutions $\text{Conv}_n(\boldsymbol{Q}, \boldsymbol{W}_{q_n})$ on the query sequence with window size $n$ to obtain the $n$-gram hidden representations of the query. Consider Figure 3, where we apply convolution on $\boldsymbol{Q}$ for $n = 1$ and $n = 2$ to generate the respective unigram and bigram queries.[1] These queries are then used to attend over unigram and bigram key-values to generate the heterogeneous attention vectors as follows.

$$\boldsymbol{A}_{1,\text{ConvKV}} = \quad \mathcal{S}(\frac{(\boldsymbol{QW}_{q_1})[(\boldsymbol{KW}_{k,1})^T; \text{Conv}_2(\boldsymbol{K}, \boldsymbol{W}_{k,2})^T; ...]}{\sqrt{d_k}})[(\boldsymbol{VW}_{v,1}); \text{Conv}_2(\boldsymbol{V}, \boldsymbol{W}_{v,2}); ...] \tag{9}$$

$$\boldsymbol{A}_{2,\text{ConvKV}} = \quad \mathcal{S}(\frac{\text{Conv}_2(\boldsymbol{Q}, \boldsymbol{W}_{q_2})[(\boldsymbol{KW}_{k,1})^T; \text{Conv}_2(\boldsymbol{K}, \boldsymbol{W}_{k,2})^T; ...]}{\sqrt{d_k}})[(\boldsymbol{VW}_{v,1}); \text{Conv}_2(\boldsymbol{V}, \boldsymbol{W}_{v,2}); ...] \tag{10}$$

$$\boldsymbol{A}_{1,\text{QueryK}} = \quad \mathcal{S}([\frac{(\boldsymbol{QW}_{q_1,1})(\boldsymbol{KW}_{k,1})^T}{\sqrt{d}}; \frac{\text{Conv}_2(\boldsymbol{KW}_{k,2}, \boldsymbol{QW}_{q_1,2})}{\sqrt{d*n_2}}; ...])[(\boldsymbol{VW}_{v,1}); \text{Conv}_2(\boldsymbol{V}, \boldsymbol{W}_{v,2}); ...] \tag{11}$$

$$\boldsymbol{A}_{2,\text{QueryK}} = \mathcal{S}([\frac{\text{Conv}_2(\boldsymbol{Q}, \boldsymbol{W}_{q_2,1})(\boldsymbol{KW}_{k,1})^T}{\sqrt{d}}; \frac{\text{Conv}_2(\boldsymbol{KW}_{k,2}, \text{Conv}_2(\boldsymbol{Q}, \boldsymbol{W}_{q_2,2}))}{\sqrt{d*n_2}}; ..])[(\boldsymbol{VW}_{v,1}); \text{Conv}_2(\boldsymbol{V}, \boldsymbol{W}_{v,2}); ..] \tag{12}$$

The result of these operations (Eq. 9 - 12) is a sequence of unigram and bigram attention vectors $\boldsymbol{A}_1 = (\mathbf{u}_1, \mathbf{u}_2, \ldots, \mathbf{u}_N)$ and $\boldsymbol{A}_2 = (\mathbf{b}_1, \mathbf{b}_2, \ldots, \mathbf{b}_{N-1})$ respectively, where $N$ is the query length. Note that each $\mathbf{u}_i \in \boldsymbol{A}_1$ represents the attention vector for $\boldsymbol{Q}_i$ unigram query, and each $\mathbf{b}_i \in \boldsymbol{A}_2$

---

[1] $\text{Conv}_1(\boldsymbol{Q}, \boldsymbol{W}_{q_1})$ is equivalent to a feed-forward connection.

represents the attention vector for $(\boldsymbol{Q}_i\text{-}\boldsymbol{Q}_{i+1})$ bigram queries, and these vectors are generated by attending to n-grams ($n = 1, 2, \ldots, N$) of keys and values.

In the next step, the phrase-level attentions in $\boldsymbol{A}_2$ are *interleaved* with the unigram attentions in $\boldsymbol{A}_1$ to form an interleaved attention sequence $\boldsymbol{I}$ such that the vectors are aligned. For unigram and bigram queries, the interleaved vector sequences at the encoder and decoder are formed as

$$\boldsymbol{I}_{\text{enc}} = (\mathbf{0}, \mathbf{u}_1, \mathbf{b}_1, \mathbf{u}_2, \mathbf{b}_2, \mathbf{u}_3, \ldots, \mathbf{b}_{N-1}, \mathbf{u}_N, \mathbf{0}) \quad (13)$$

$$\{\boldsymbol{I}_{\text{dec}}, \boldsymbol{I}_{\text{cross}}\} = (\mathbf{0}, \mathbf{u}_1, \mathbf{b}_1, \mathbf{u}_2, \mathbf{b}_2, \mathbf{u}_3, \ldots, \mathbf{b}_{N-1}, \mathbf{u}_N) \quad (14)$$

where $\boldsymbol{I}_{\text{enc}}$ and $\boldsymbol{I}_{\text{dec}}$ denote the interleaved sequence for self-attention at the encoder and decoder respectively, and $\boldsymbol{I}_{\text{cross}}$ denotes the interleaved sequence for cross-attention between the encoder and the decoder. Note that although $\boldsymbol{I}_{\text{dec}}$ and $\boldsymbol{I}_{\text{cross}}$ are computed using the same formula (Eq. 14), they are different entities, operating over different input sequences – the input to $\boldsymbol{I}_{\text{dec}}$ comes from the self-attended features in the target side, whereas the input to $\boldsymbol{I}_{\text{cross}}$ comes from the cross-attended features from source. Also, to prevent information flow from the future in the decoder, the right connections are masked out in $\boldsymbol{I}_{\text{dec}}$ and $\boldsymbol{I}_{\text{cross}}$ (similar to the original Transformer). The interleaving operation places the phrase- and token-based representations of a token next to each other. The interleaved vectors are finally passed through a convolution layer (as opposed to a point-wise feed-forward layer in the Transformer) to compute the overall representation for each token. By doing so, each query is intertwined with the n-gram representations of the phrases containing itself, which enables the model to learn the query's correlation with neighboring tokens. For unigram and bigram queries, the encoder uses a convolution layer with a window size of 3 and stride of 2 to allow the token to intertwine with its past and future phrase representations, while the ones in the decoder (self- and cross-attention) use a window size of 2 and stride of 2 to incorporate only the past phrase representations to preserve the autoregressive property. More formally,

$$\boldsymbol{O}_{\text{enc}} = \text{Conv}_{\text{window=3,stride=2}}(\boldsymbol{I}_{\text{enc}}, \boldsymbol{W}_{\text{enc}}) \quad (15)$$

$$\boldsymbol{O}_{\text{cross}} = \text{Conv}_{\text{window=2,stride=2}}(\boldsymbol{I}_{\text{cross}}, \boldsymbol{W}_{\text{cross}}) \quad (16)$$

$$\boldsymbol{O}_{\text{dec}} = \text{Conv}_{\text{window=2,stride=2}}(\boldsymbol{I}_{\text{dec}}, \boldsymbol{W}_{\text{dec}}) \quad (17)$$

## 4 EXPERIMENTS

In this section, we present the training settings, experimental results and analysis of our models.

### 4.1 TRAINING SETTINGS

We preserve most of the training settings from Vaswani et al. (2017) to enable a fair comparison with the original Transformer. Specifically, we use the Adam optimizer (Kingma & Ba, 2014) with $\beta_1 = 0.9$, $\beta_2 = 0.98$, and $\epsilon = 10^{-9}$. We follow a similar learning rate schedule with $warmup\_steps$ of 16000: $LearningRate = 2 * d^{-0.5} * \min(step\_num^{-0.5}, step\_num * warmup\_steps^{-1.5})$. While Vaswani et al. (2017) trained their base and big models at a massive scale with 8 GPUs, we could train our models only on a single GPU because of limited GPU facilities. We trained our models and the baseline Transformer on the same GPU for 500,000 steps. The batches were formed by sentence pairs containing approximately 4096 source and 4096 target tokens. Similar to Vaswani et al. (2017), we also applied residual dropout with 0.1 probability and label smoothing with $\epsilon_{ls} = 0.1$. Our models are implemented in the *tensor2tensor*[2] library (Vaswani et al., 2018), on top of the original Transformer codebase. We conducted all the experiments with our models and the original Transformer in an identical setup for a fair comparison.

We trained our models on the standard WMT'16 English-German (En-De) and English-Russian (En-Ru) datasets constaining about 4.5 and 25 million sentence pairs, respectively. We used WMT **newstest2013** as our development sets and **newstest2014** as our test sets for all the translation tasks. We used byte-pair encoding (Sennrich et al., 2016) with combined source and target vocabulary of 37,000 sub-words for English-German and 40,000 sub-words for English-Russian. We took the average of the last 5 checkpoints (saved at 10,000-iteration intervals) for evaluation, and used a beam search size of 5 and length penalty of 0.6 (Wu et al., 2016).

---

[2]https://github.com/tensorflow/tensor2tensor

| Model | Technique | # Parameters | N-grams | En→De | De→En | En→Ru | Ru→En |
|---|---|---|---|---|---|---|---|
| Transformer big | - | 214M | - | 26.63 | — | 34.64 | — |
| Transformer base | - | 63M | - | 26.07 | 29.82 | 35.64 | 34.56 |
| Homogeneous | CONVKV | 67M | 4/4 | 26.60 | 30.17 | 36.14 | 34.75 |
| Homogeneous | CONVKV | 74M | 3/2/3 | 26.55 | 30.03 | 36.31 | 34.65 |
| Homogeneous | QUERYK | 67M | 4/4 | 26.78 | 30.03 | 36.01 | 34.35 |
| Homogeneous | QUERYK | 74M | 3/2/3 | 26.86 | 29.87 | 36.10 | 34.59 |
| Heterogeneous | CONVKV | 81M | 1-2 | 27.04 | 30.09 | 36.83 | 35.10 |
| Heterogeneous | CONVKV | 110M | 1-2-3 | 27.15 | 30.29 | 36.90 | 35.68 |
| Heterogeneous | CONVKV | 148M | 1-2-3-4 | 27.15 | 30.09 | 37.28 | **35.91** |
| Heterogeneous | QUERYK | 81M | 1-2 | 26.95 | 30.20 | 36.75 | 35.01 |
| Heterogeneous | QUERYK | 110M | 1-2-3 | 27.09 | 30.29 | 37.16 | 35.17 |
| Heterogeneous | QUERYK | 148M | 1-2-3-4 | 27.08 | 30.21 | **37.39** | 35.43 |
| Interleaved | CONVKV | 116M | 1-2 | 27.33 | 30.17 | 37.16 | 34.70 |
| Interleaved | QUERYK | 116M | 1-2 | **27.40** | **30.30** | 37.24 | 34.60 |

Table 1: BLEU (cased) scores on WMT'14 testsets for English-German (**En-De**) and English-Russian (**En-Ru**) language pairs (in both directions). All models were trained with **1 GPU**. The **# Parameters** is shown in approximate terms. For homogeneous models, the **N-grams** denote how we distribute the 8 heads to different n-gram types; *e.g.,* '3/2/3' means 3 heads on unigrams, 2 on bigrams and 3 on trigrams. For heterogeneous, the numbers indicate the phrase lengths of the collection of n-gram components jointly attended by each head; *e.g.,* '1-2' means attention scores are computed across unigram and bigram logits.

## 4.2 RESULTS

Table 1 compares our model variants with the Transformer base and big models for En-De and En-Ru translation tasks (both directions). We notice that almost all of our models achieve higher BLEU scores than the Transformer base, showing the effectiveness of our approach.

On the **En→De** translation task, our best **homogeneous** model (QUERYK with 3/2/3 head distribution) achieves a BLEU of 26.86, already outperforming the Transformer base by about 0.8 points. When we compare the results of the homogeneous models with those of the **heterogeneous**, we notice even higher scores for the heterogeneous models; the best heterogeneous model yields a BLEU of 27.15, which is about 1.1 points higher than the Transformer base. Finally, we notice that our **interleaved** heterogeneous models surpass all aforementioned scores achieving up to 27.4 BLEU and establishing a 1.33 BLEU improvement over the Transformer base. This demonstrates the existence of phrase-to-token and phrase-to-phrase mappings from target (De) to source (En) language.

Likewise, on **De→En**, our models achieve improvements compared to the Transformer, but the gain is not as high as in En→De. Specifically, homogeneous and heterogeneous attentions perform comparably, giving up to +0.38 BLEU improvements compared to the Transformer base. Our interleaved models show more improvements (up to 30.3 BLEU), outperforming the Transformer by about 0.5 points. This again demonstrates the importance of phrase-level query representation in the target.

Similarly, on the **En→Ru** translation task, all of our models surpass the Transformer base. Homogeneous models with CONVKV perform slightly better than their QUERYK counterparts. In the heterogeneous family, all of the model variants experimented with achieved higher score than the homogeneous models. The QUERYK heterogeneous model with 1-2-3-4 N-grams achieves the highest performance with **37.39** BLEU, 1.75 points higher than the Transformer base. Interleaved homogeneous models with 1-2 N-grams also perform well on this task with a BLEU of 37.24, which is more than the heterogeneous models with 1-2-3 N-grams, having similar number of parameters.

On the **Ru→En** task, homogeneous models perform slightly better than the Transformer base, while heterogeneous models excel in performance. In particular, CONVKV heterogeneous model with 1-2-3-4 N-grams achieves 35.91 BLEU, which outperforms the Transformer base by 1.35 BLEU points. However, interleaved attention models do not provide significant improvements on this task. We suspect this is because the morphological richness of Russian is not fully leveraged in the current interleaved models as they are limited to only 1-2 N-grams. We put this issue in our future work.

| Model | GPUs | Batch | Steps | # Layers | # Parameters | N-grams | En→De |
|---|---|---|---|---|---|---|---|
| Transformer base Vaswani et al. (2017) | 8 | 25000 | 100K | 6 | 63M | - | 27.30 |
| Transformer big Vaswani et al. (2017) | 8 | 25000 | 100K | 6 | 214M | - | 28.40 |
| Relative Position base Shaw et al. (2018) | 8 | 25000 | 100K | 6 | - | - | 26.80 |
| Weighted Transformer large Ahmed et al. (2017) | 8 | 25000 | 60K-120K | 6 | 214M | - | 28.90 |
| Transformer big | 1 | 2048 | 1M | 6 | 214M | - | 26.63 |
| Transformer base | 1 | 4096 | 500K | 10 | 91M | - | 26.60 |
| Transformer base | 1 | 4096 | 500K | 6 | 63M | - | 26.07 |
| Heterogeneous (ConvKV) | 1 | 4096 | 500K | 4 | 60M | 1-2 | 26.63 |
| Heterogeneous (ConvKV) | 1 | 4096 | 500K | 6 | 81M | 1-2 | 27.04 |
| Heterogeneous (QueryK) | 1 | 4096 | 500K | 6 | 81M | 1-2 | 26.95 |
| Interleaved (QueryK) | 1 | 4096 | 500K | 6 | 116M | 1-2 | **27.40** |

Table 2: BLEU scores on WMT'14 **English-to-German** translation task for different models with different training settings. The first block presents **reported results** along with the **number of GPUs**, **batch size**, and **number of update steps** used from the respective papers. The second and third block show results in our training setting with 1 GPU.

**Homogeneous vs. Heterogeneous.** Heterogeneous models generally perform better than their homogeneous counterparts. This shows the effectiveness of relaxing the 'same n-gram type' attention constraint in the heterogeneous head, which allows it to attend to all n-gram types simultaneously, avoiding any forced attention to a particular n-gram type. In fact, our experiments show that the models perform worse than the baseline if we remove token-level attentions.

**Effect of Higher-Order N-grams.** We now analyze the effect of including higher-order n-grams. For homogeneous models, including 3-gram components does not have any significant effect in performance, while it increases the number of parameters. On the other hand, heterogeneous models benefit significantly from 3-gram and 4-gram components. For **En→De**, 3-gram and 4-gram components offer 0.1 - 0.2 improvements in BLEU. Meanwhile, for **En → Ru** and **Ru → En**, they yield up to 0.64 BLEU improvements compared to the model using only uni- and bi-grams. This supports the argument that the usefulness of 3-grams and 4-grams depends on the language pair, and our heterogeneous models are more robust in selecting any type of n-grams that are useful.[3]

**Comparison with State-of-the-art.** Table 2 places our BLEU scores with the state-of-the-art results **reported** on the **En→De** translation task. Note that Vaswani et al. (2017) and others conducted their experiments at a massive scale with 8 GPUs, which we could not replicate due to limited number of GPUs. Therefore, we conducted all experiments using a single GPU, and make comparisons in this 'low-resource' setting. There have been many evidences that practical training of the Transformer networks (theirs and ours) is significantly susceptible to the batch size (which increases with the number of GPUs used), and training on a single GPU with lower batch size for sufficiently long does not produce similar results as in 8 or more GPU settings; see (Popel & Bojar, 2018) for details.[4]

As can be noticed, in our training setting (1 GPU), the Transformer big performs only slightly better than the Transformer base, giving 26.63 BLEU in contrast to the reported 28.40 BLEU with 8 GPUs. We believe, the difference in performance is due to the batch size, for which we could conduct trials only at 2048 tokens instead of 25000, although we trained the big model 10 times longer.

In order to verify if phrasal attention (which adds an additional convolution layer in parallel to the original Transformer layer), does indeed help more than just placing extra Transformer layers, we conducted additional experiments with the Transformer base with 10 layers, and with our heterogeneous model with 4 layers. We notice that our heterogeneous model with 6 layers outperforms the Transformer base with 10 layers and the Transformer big, and with 4 layers, it performs on par, even though it has much less parameters. These differences are even more significant for **En→Ru**, where the Transformer big underperforms the Transformer base with a BLEU of 34.64 (Table 1). On the other hand, our heterogeneous model achieves 37.39 BLEU (+ 2.75). Therefore, our phrasal attention can achieve similar or higher results with a wider but shallower network that consumes less parameters compared to deep Transformer base or Transformer big. This increases parallelizability.

---

[3]It is nontrivial to include higher-order n-grams (*i.e.,* n>2) in the interleaved model, because one needs to mix the sequences into one such that different n-gram types align properly. We leave it to future work.

[4]Also see the discussion https://github.com/tensorflow/tensor2tensor/issues/444

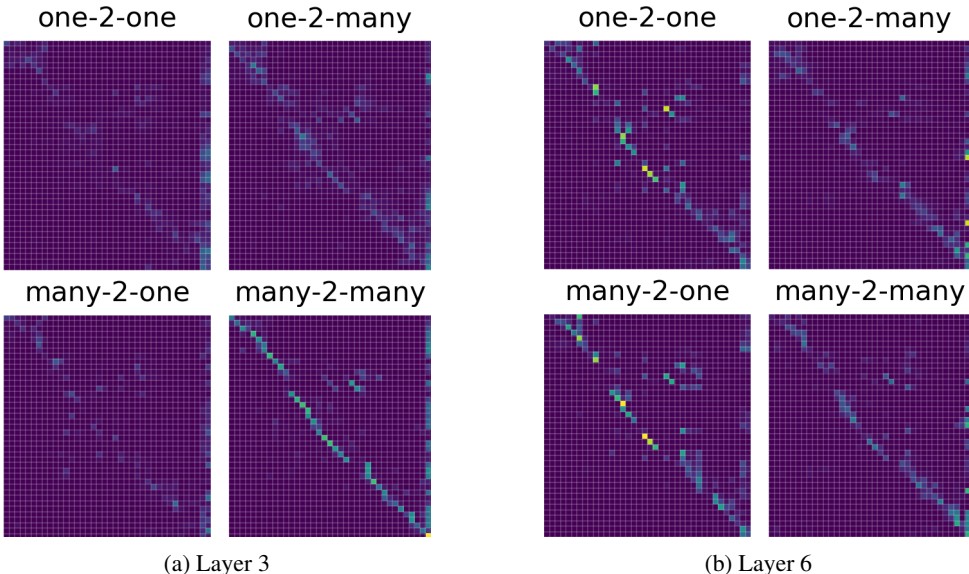

(a) Layer 3                          (b) Layer 6

Figure 4: Sample attention heat map of QUERYK interleaved heterogeneous model.

Phrasal attentions are also more suitable for 'low-resource' (GPU) setup, and for complex language pairs like English-Russian, where our methods show larger improvements.

**Model Interpretation.** To interpret our phrasal attention models, we now discuss how they learn the alignments. Figure 4 shows attention heatmaps for an En→De sample in newstest2014; figure 4a displays the heatmap in layer 3 (mid layer), while figure 4b shows the one in layer 6 (top layer) within a 6-layer Transformer based on our interleaved attention. Each figure shows 4 quadrants representing *token-to-token, token-to-phrase, phrase-to-token, phrase-to-phrase* attentions, respectively. We can see in this example that phrasal attentions are activated strongly in the mid-layers; particularly, the phrase-to-phrase attention is the most concentrated one. On the other hand, token-to-token attention is activated the most in the top layer. Although the distribution of attentions can be quite different depending on the model initialization, we observed a large portion of the attentions for phrases, as shown in Tables 3 and 4 in the Appendix for two different random seeds.

## 5 CONCLUSIONS

We have presented novel approaches to incorporating phrasal alignments into the attention mechanism of state-of-the-art neural machine translation models. Our methods assign attentions to all four possible mapping relations between target and source sequences: token-to-token, token-to-phrase, phrase-to-phrase and phrase-to-phrase. While we have applied our attention mechanisms to the Transformer network, they are generic and can be implemented in other architectures. On WMT'14 English-German translation tasks, all of our methods surpass the Transformer base model. Our model with interleaved heterogeneous attention, which tackles all the possible phrasal mappings in a unified framework, achieves improvements of 1.33 BLEU for English-to-German and 0.48 BLEU for German-to-English translation tasks over the baseline model. Likewise for WMT'14 English-Russian translation tasks, our models achieve up to 1.75 BLEU for English-to-Russian and 1.35 BLEU for Russian-to-English translation compared to the baseline. We are planning future extensions of our techniques to other tasks, such as summarization and question answering. We also plan to improve our models with a phrase-based decoding procedure.

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

# 6 APPENDIX

## 6.1 ATTENTION HEAT MAP OF QUERYK HETEROGENEOUS MODEL

(a) Layer 3 of QUERYK heterogeneous

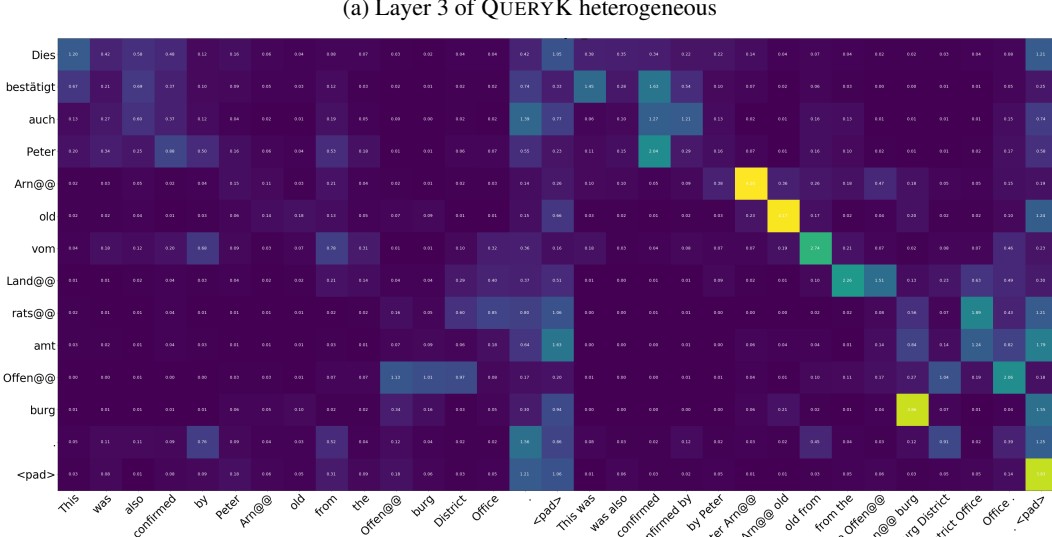

(b) Layer 6 QUERYK heterogeneous

Figure 5: Attention heat maps at layer 3 and layer 6 of **QUERYK heterogeneous** model for a sample sentence pair in English-German newstest2014 test set. The left half in each figure indicates **token-to-token** mappings, while the right half indicates **token-to-phrase** mappings.

## 6.2 ATTENTION STATISTICS FOR INTERLEAVED HETEROGENEOUS MODEL

Tables 3 and 4 show the distributions of phrase- and token-based attentions across different layers for two different random seeds. Depending on the initial states, the allocations can be different, but phrasal attentions always play an important role in learning the mapping from source to target.

## 6.3 ATTENTION HEAT MAP OF QUERYK INTERLEAVED MODEL

| Layer | token-to-token | token-to-phrase | phrase-to-token | phrase-to-phrase |
|-------|----------------|-----------------|-----------------|------------------|
| 1 | 98.01 | 00.20 | 01.60 | 00.19 |
| 2 | 39.63 | 19.49 | 02.73 | 38.15 |
| 3 | 01.54 | 02.30 | 00.00 | 96.16 |
| 4 | 36.13 | 37.53 | 06.60 | 19.74 |
| 5 | 61.96 | 18.12 | 08.59 | 11.33 |
| 6 | 53.72 | 10.53 | 34.63 | 01.12 |

Table 3: For model trained with seed 100, percentage (%) of activations for different attention types in each layer of the Interleaved model for English-to-German translation task in newstest2014.

| Layer | token-to-token | token-to-phrase | phrase-to-token | phrase-to-phrase |
|-------|----------------|-----------------|-----------------|------------------|
| 1 | 17.50 | 16.14 | 66.34 | 00.03 |
| 2 | 00.47 | 98.60 | 00.02 | 00.91 |
| 3 | 02.28 | 45.34 | 00.01 | 54.35 |
| 4 | 08.68 | 02.28 | 00.00 | 89.04 |
| 5 | 95.32 | 00.58 | 00.16 | 03.95 |
| 6 | 14.70 | 27.01 | 57.41 | 00.80 |

Table 4: For model trained with seed 20, percentage (%) of activations for different attention types in each layer of the Interleaved model for English-to-German translation task in newstest2014.

(a) Layer 3 of QUERYK interleaved

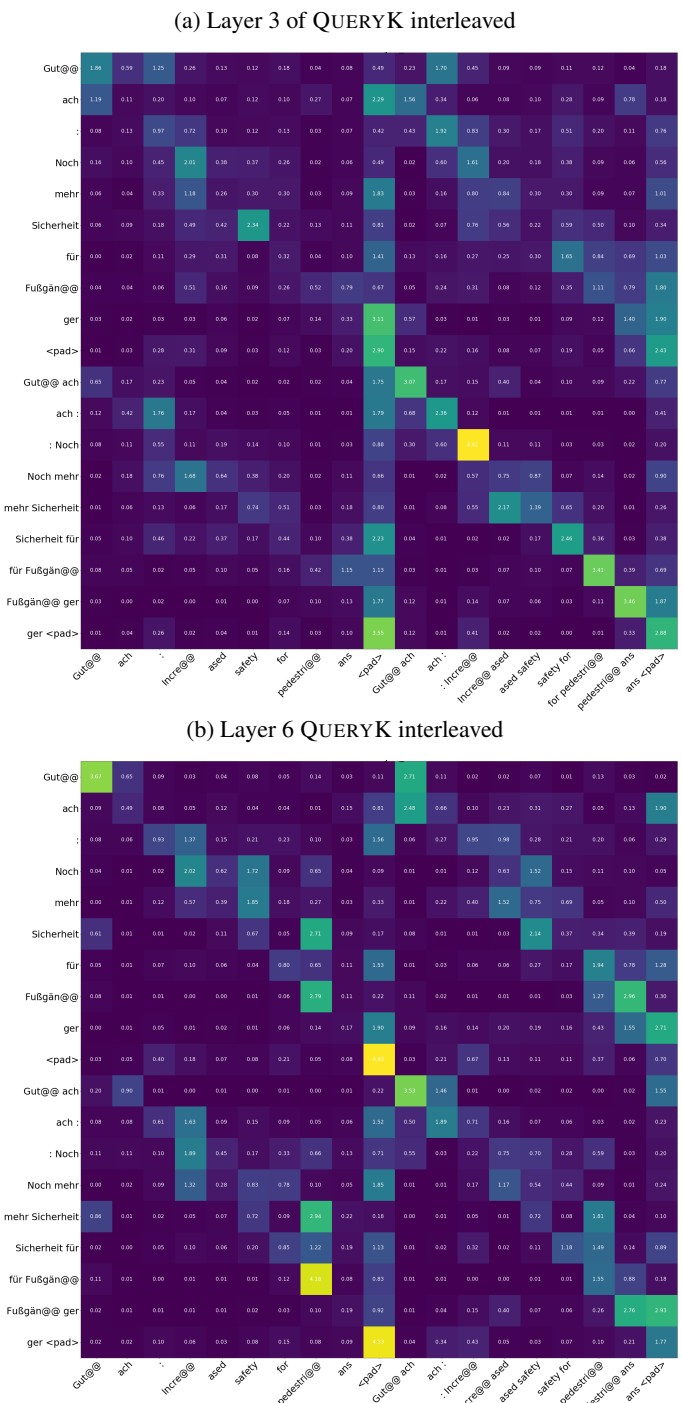

(b) Layer 6 QUERYK interleaved

Figure 6: Attention heat maps at layer 3 and layer 6 of **QUERYK interleaved heterogeneous** model for another sample from English-German newstest2014 test set. Upper-left, upper-right, lower-left, lower-right quadrants of each figure show token-to-token, token-to-phase, phrase-to-token, phrase-to-phrase alignments respectively.

