# OpenReview forum: "Phrase-Based Attentions"
_ICLR.cc/2019/Conference_

### Official Review · AnonReviewer2 · 2018-10-31
**Potentially useful extension of the Transformer model, but needs more solid experiments.**

**Rating:** 5
**Confidence:** 5

**Review:**

The authors propose to include phrases (contiguous n-grams of wordpieces) in both the self-attention and encoder-decoder attention modules of the Transformer model (Vaswani et al., 2017). In standard multi-head attention, the logits of the attention distribution of each head is computed as the dot-product between query and key representations, which are position-specific. In the phrase-based attention proposed here, a convolution is first computed over the query, key, value sequences before logits are computed as before (a few variants of this scheme are explored). Results show an improvement of up to 1.3 BLEU points compared to the baseline Transformer model. However, the lack of a controlled experiment sheds substantial doubt on the efficiacy of the model (see below).

Contributions
-------------------
Proposes a simple way to incorporate n-grams in the Transformer model. The implementation is straightforward and should be fully replicable in an afternoon.

Having an inductive bias towards modeling of longer phrases seems intuitively useful, in particular when using subword representations, where subword units are often ambiguous. This is also motivated by the fact that prior work has shown that subword regularization, where sampling different subword segmentations during training can be useful.

Improvements in BLEU scores are quite strong.

Issues
---------
The experiments do not control for parameter count. The phrasal attention model adds significant number of parameters (e.g., "interleaved attention" corresponds to 3x the number of 1D convolution parameters in the attention layer). It is well established that more parameters correspond to increased BLEU scores (e.g., the 2x parameter count in the "big" Transformer setting from Vaswani et al. (2017) results in over 1 BLEU point improvement). This needs to be fixed!

The model is a very modest extension of the original Transformer model and so its value to the community beyond improved numbers is somewhat questionable.

While an explicit inductive bias for phrases seem plausible, it may be that this can already be fully captured by multi-head attention. With positional encodings, two heads can easily attend to adjacent positions which gives the model the same capacity as the convolutional phrase model. The result in the paper that trigrams do not add anything on top of bigrams signals to me that the model is already implicitly capturing phrase-level aspects in the multi-head attention. I would urge the authors to verify this by looking at gradient information (https://arxiv.org/abs/1312.6034).

There are several unsubstantiated claims: "Without specific attention to phrases, a particular attention function has to depend entirely on the token-level softmax scores of a phrase for phrasal alignment, which is not robust and reliable, thus making it more difficult to learn the mappings." - The attention is positional, but not necessarily token-based. The model has capacity to represent phrases in subsequent layers. WIth h heads , a position in the k-th layer can in principle represent h^k grams (each slot in layer 2 can represent a h-gram and so on).

The differences in training setup compared to Vaswani et al. (2017) needs to be explicit ("most of the training settings" is too handwavy). Please list any differences.

The notation is somewhat cumbersome and could use some polishing. For example, the input and output symbols both range over indices in [1,n]. The multi-head attention formulas also do not match the ones from Vaswani et al. (2017) fully. Please ensure consistency and readability of symbols and formulas.

The model inspection would be much improved by variance analysis. For example, the numbers in table 3 would be more useful if accompanied by variance across training runs. The particular allocation could well be an effect of random initialization. I could also see other reasons for this particular allocation than phrases being more useful in intermediate layers (e.g., positional encodings in the first layer is a strong bias towards token-to-token attention, it could be that the magnitude of convolved vectors is larger than the batch-normalized unigram encodings, so that logits are larger.

Questions
--------------
In "query-as-kernel convolution", it is unclear whether you map Q[t, :] into n x d_q x d_k convolution kernel parameters, or if each element of the window around Q[t] of width n is mapped to a convolution kernel parameter. Also what is the exact form of the transformation. Do you transform the d_q dimensional vectors in Q to a d_q x d_k matrix? Is this done by mapping to a d_q * d_k dimensional vector which is then rearranged into the convolution kernel matrix?

Does the model tend to choose one particular n-gram type for a particular position, or will it select different n-gram types for the same position?

"The selection of which n-gram to assign to how many heads is arbitrary" - How is this arbitrary? This seems a rather strong inductive bias?

"However, the homogeneity restriction may limit the model to learn interactions between different n-gram types" - How is the case? It seems rather that the limitation is that the model cannot dynamically allocate heads to the most relevant n-gram type?

I do not understand equation 14. Do you mean I_dec = I_cross = (...)?

"Phrase-to-phrase mapping helps model local agreement, e.g., between an adjective and a noun (in terms of gender, number and case) or between subject and verb (in terms of person and number)." Is this actually verified with experiments / model inspection?

"This is especially necessary when the target language is morphologically rich, like German, whose words are usually compounded with sub-words expressing different meanings and grammatical structures" This claim should be verified, e.g. by comparing to English-French as well as model inspection.

---

> ### Author Response · Authors · 2018-11-13
> **Response to reviewer's comment**
>
> Thank you for your insightful reviews. We address your comments as follows.
>
> 1/ The experiments do not control for parameter count. The phrasal attention model adds significant number of parameters (e.g., "interleaved attention" corresponds to 3x the number of 1D convolution parameters in the attention layer). It is well established that more parameters correspond to increased BLEU scores (e.g., the 2x parameter count in the "big" Transformer setting from Vaswani et al. (2017) results in over 1 BLEU point improvement). This needs to be fixed!
>
> We have now provided parameter counts of the models with the results in Table 1 in the revised version of the paper. Thanks for the suggestion. However, with due respect, the parameter counts do not match with your estimates; the base transformer has 63M parameters, the interleaved attention (our biggest model) has 116M parameters, while the transformer big has 214M parameters. That is, our interleaved model has approximately 1.8x more parameters than the transformer base, while the transformer big has almost 3.4x more parameters than the base transformer.
> In addition, what really makes the difference in BLEU scores between Vaswani et al. (2017) and ours is the batch size (or the number of GPUs) used for training, which has been shown empirically to affect the results substantially. Please see the discussion  https://github.com/tensorflow/tensor2tensor/issues/444 or this material https://ufal.mff.cuni.cz/pbml/110/art-popel-bojar.pdf for more details.. We also observed the same issue for both the transformer and our phrasal attentions. The transformer used a massive setting of 8 GPUs, while we could only afford to run our experiments on a single GPU. Therefore, we compare our models with the base transformer that is trained with 1 GPU using the same settings and codes provided by the authors. This baseline gives 26.07 BLEU. We claim our gain (1.3 BLEU) from this baseline instead of the reported 8-GPU result to ensure a fair comparison.
>
> Having said that, we conducted more experiments on English - Russian and Russian - English and conduct analysis and experiments on the number of parameters relative to base transformer and big transformer.
>
> For En-Ru:
> Base transformer:                                                         35.64
> Homogeneous QueryKernel:                                        36.31
> Heterogeneous ConvKV with n-gram (1-2):                 36.81
> Heterogenenous QueryKernel with n-gram (1-2-3-4):  37.39
> Interleaved heterogeneous QueryKernel (1-2):            37.24
>
> For Ru-En
> Base transformer:                                                         34.56
> Homogeneous ConKV:                                                 34.75
> Heterogenenous ConvKV with n-gram (1-2):               35.10
> Heterogenenous ConvKV with n-gram (1-2-3-4):         35.91
> Interleaved heterogeneous ConvKV (1-2):                   34.70
>
> We also conducted more experiments as per your request:
> transformer base  (6 layers):            63M params,        26.08 BLEU
> transformer big (6 layers):                214M params,      26.63 BLEU
> transformer base (10 layers):           91M params,        26.60 BLEU
> heterogeneous 4 layers (ConvKV):  60M params,        26.63 BLEU
> heterogeneous 6 layers (ConvKV):  80M params,        27.04 BLEU
>
> As you may see, transformer big and transformer base (10-layers) have more parameters; they should perform better than transformer base. But the margins are not significant because of limited batch size (explained later). On the other hand, our heterogeneous model with just 4 layers is already on par with transformer base 10 layers or transformer big, though it has less parameters.
>
> Response continues in the next comment.

---

> > ### Author Response · Authors · 2018-11-13
> > **Continued response**
> >
> > 2/ The model is a very modest extension of the original Transformer model and so its value to the community beyond improved numbers is somewhat questionable.
> >
> > Our main contributions in this paper are:
> >
> > Propose two new computational methods to be used to attend to phrases instead of tokens.
> > Key-Value Convolution
> > Query-as-Kernel  Convolution
> > Propose three different ways to incorporate these phrasal attention into the existing token-level attention
> > Homogeneous Attention: where we select each head in the multi-head group to attend on different n-gram types separately (local Softmax).
> > Heterogenous Attention: where we jointly attend phrases alongside unigrams in every head with a global Softmax.
> > Interleaved Heterogeneous Attention: This is a considerable extension to heterogeneous approach as it incorporates phrases from the target by performing phrasal queries.
> >
> > These methods give a complete solution to a novel idea of phrase-based attentions for NMT. We firmly believe, this is not just a very modest extension to the original transformer, especially the three ways to incorporate phrasal attentions in the transformer. We hope the reviewer also considers them as significant contributions.
> >
> > 3/ While an explicit inductive bias for phrases seem plausible, it may be that this can already be fully captured by multi-head attention. With positional encodings, two heads can easily attend to adjacent positions which gives the model the same capacity as the convolutional phrase model. The result in the paper that trigrams do not add anything on top of bigrams signals to me that the model is already implicitly capturing phrase-level aspects in the multi-head attention. I would urge the authors to verify this by looking at gradient information (https://arxiv.org/abs/1312.6034).
> >
> > Good comment! We agree that in theory, the original token-based attention can attend to adjacent positions in a sequence. Even without multiple heads, attention can do so by putting equally higher weights to the associated tokens of the bigrams or trigrams. But there are doubts if it really does so, and there is no such evidence suggesting that it does. The same is true for multi-head attentions; it can in theory attend to adjacent positions, but whether it really does so and does it correctly remains questionable. On the other hand, empirical results suggest otherwise. For example, the attention heat maps of (Vaswani et al. 2017) and ours show that the attention tends to concentrate the scores on individual tokens even if it uses multiple heads concurrently. This findings motivated us to propose specialized mechanisms for the transformer to attend to phrases explicitly and effectively.
> >
> > The finding that trigrams do not add anything significant on top of bigrams happens only with the homogeneous architecture for En-De language pair. However, trigrams do indeed yield better performance with the heterogeneous approach (as seen in Table 2). Especially, in our new experiments with the En - Ru language pair, we see consistent gains in both translation directions for including higher-order n-grams (please see our revised paper). Therefore, we believe, this depends on the language pairs. We have revised our paper accordingly with the new findings on En-Ru.
> > As mentioned in the paper, tor the homogeneous model, trigrams are forced to be attended even when it is not needed, which imposes noise to the model. For the heterogeneous model, trigram attentions are optional so it will leverage them only when there is a need to do so depending on the input-output pair.
> >
> > Response continues in the next comment.

---

> > > ### Author Response · Authors · 2018-11-13
> > > **Continued response**
> > >
> > > 4/ The differences in training setup compared to Vaswani et al. (2017) needs to be explicit ("most of the training settings" is too handwavy). Please list any differences.
> > >
> > >
> > > We have explicitly stated the differences in the revised version. The only differences are the number of GPUs used (1 instead of 8) and the number of training steps (500000 instead of 100000).
> > >
> > > 5/ The notation is somewhat cumbersome and could use some polishing. For example, the input and output symbols both range over indices in [1,n]. The multi-head attention formulas also do not match the ones from Vaswani et al. (2017) fully. Please ensure consistency and readability of symbols and formulas.
> > >
> > >
> > > We have fixed the input/output indices in the revised version (Sec. 2).
> > > We think, the stated multi-head attention formula in our paper is a clearer version of the one in Vaswani et al. (2017). In their paper, Q and K and V are actually passed through a element-wise dense layer (1x1 Conv) with weight W before going through the main attention function: softmax(QK^T / sqrt(d_model))V.
> > > In our paper, we reiterate that formula by putting those element-wise dense layer into the formula, because this makes the difference in the formula of transformer and phrasal attention and we wanted to emphasize that. Therefore, the condensed formula is softmax((QW_q) (KW_k)^T / sqrt(d_model)) (VW_v).
> > > Another difference is that we used the symbol S to represent softmax. The reason is that the equations 9-12 are too long to fit nicely in the paper layout. So to be consistent, we used S everywhere.
> > >
> > >
> > > 6/ The model inspection would be much improved by variance analysis. For example, the numbers in table 3 would be more useful if accompanied by variance across training runs. The particular allocation could well be an effect of random initialization. I could also see other reasons for this particular allocation than phrases being more useful in intermediate layers (e.g., positional encodings in the first layer is a strong bias towards token-to-token attention, it could be that the magnitude of convolved vectors is larger than the batch-normalized unigram encodings, so that logits are larger.
> > >
> > >
> > > We have experimented with a different random seed and you are right - the attention allocations are indeed an effect of the random initialization. We get very different heatmaps as shown below.
> > >
> > > REPORT THE NUMBERS HERE
> > >
> > > % Seed 100
> > > % BLEU: 27.32
> > >
> > >   T-to-T    T-to-P   P-to-T   P-to-P
> > > Layer 1  [ 98.012   0.202     1.597   0.189]
> > > Layer 2  [ 39.63     19.487   2.727   38.156]
> > > Layer 3  [   1.544   2.297     0.         96.159]
> > > Layer 4  [ 37.118   37.516   6.591   18.775]
> > > Layer 5  [ 61.959   18.12     8.589   11.332]
> > > Layer 6  [ 53.72     10.532   34.634  1.115]
> > >
> > > % Seed 20
> > > % BLEU: 27.33
> > >
> > > 	T-to-T    T-to-P   P-to-T   P-to-P
> > > Layer 1  [17.495 16.141  66.336  0.028]
> > > Layer 2  [ 0.469  98.603  0.021    0.907]
> > > Layer 3  [ 2.283  45.358  0.007    52.351]
> > > Layer 4  [ 8.679  2.283    0.          89.039]
> > > Layer 5  [95.316 0.584    0.155    3.946]
> > > Layer 6  [14.699 27.091  57.41    0.8  ]
> > >
> > > We have revised our paper accordingly. Regarding the magnitude of convolved vectors, the scaling factor sqrt(d_model * n) (Eq. 6) should take care of this. Nevertheless, our new run with a different seed shows that the attentions over tokens and phrases do not have any distinct patterns, rather they are distributed across the layers. Sorry about the confusion; we were misled by the nice patterns in the initial run. Thanks for your valuable suggestion.
> > >
> > > Response continues in the next comment.

---

> > > > ### Author Response · Authors · 2018-11-13
> > > > **Continued response**
> > > >
> > > > 7/ In "query-as-kernel convolution", it is unclear whether you map Q[t, :] into n x d_q x d_k convolution kernel parameters, or if each element of the window around Q[t] of width n is mapped to a convolution kernel parameter. Also what is the exact form of the transformation. Do you transform the d_q dimensional vectors in Q to a d_q x d_k matrix? Is this done by mapping to a d_q * d_k dimensional vector which is then rearranged into the convolution kernel matrix?
> > > >
> > > >
> > > > We map Q[t, :] of dimension t x d_q to t x n*d_k using an 1x1 convolution (or element-wise dense layer) with a weight matrix of d_q x n*d_k dimensions. Then we reshape the result to t x n x d_k x 1 to make each q in Q[t, :] become a convolutional filter.
> > > >
> > > > 8/ Does the model tend to choose one particular n-gram type for a particular position, or will it select different n-gram types for the same position?
> > > >
> > > >
> > > > As described in the paper, in homogeneous attention models, each head is forced to choose one single n-gram type that the head represents. In heterogeneous attention models, the head can freely choose any of the n-gram types. As illustrated in the heatmaps (fig 4, fig 5, fig 6) and the heat map percentage tables (table 3, table 4), the model tends to attend different n-gram types based on what the query is, the locations, and the layer within the model. To answer your question, it selects different n-gram types for the same position. These are our observations based on the heat map images and the statistical analysis.
> > > >
> > > > 9/ "The selection of which n-gram to assign to how many heads is arbitrary" - How is this arbitrary? This seems a rather strong inductive bias?
> > > > What we mean here is that this is a hyperparameter for the homogeneous model that one should tune for the best outcome. For example, given a 8-head attention module, one can assign 4-4 (4 unigram, 4 bigram heads), 3-2-3 (3 unigram, 2 bigram and 3 trigram heads), 2-2-2-2, or 1-2-3-2.
> > > >
> > > > 10/ "However, the homogeneity restriction may limit the model to learn interactions between different n-gram types" - How is the case? It seems rather that the limitation is that the model cannot dynamically allocate heads to the most relevant n-gram type?
> > > >
> > > >
> > > > What we mean here is that the heads in homogeneous models attend on different n-gram types separately and independently, and the attention results are merged by concatenation to it pass to the next layer. So there is no cross interaction between the n-gram types. The gradients flow through them in parallel paths with no interactions.
> > > > It is not that the model cannot dynamically allocate to the most relevant n-grams, but rather it is forced to allocate to one type even though there might not be any useful and relevant n-grams.
> > > >
> > > > 11/ I do not understand equation 14. Do you mean I_dec = I_cross = (...)?
> > > >
> > > >
> > > > Sorry for the confusion! What we mean here is that I_dec and I_cross are computed using the same formula, which is  different from how I_enc is computed. I_dec != I_cross, because those are 2 different things and their inputs are different -- the input (u1, b1,...) to I_dec is not the same as the input (u1, b1,...) to I_cross.
> > > >
> > > > 12 / "Phrase-to-phrase mapping helps model local agreement, e.g., between an adjective and a noun (in terms of gender, number and case) or between subject and verb (in terms of person and number)." Is this actually verified with experiments / model inspection?
> > > >
> > > >
> > > > These are the advantages of phrase-based SMT over word-based SMT. We believe the same advantages hold for NTM with phrasal attentions.
> > > >
> > > > 13/ "This is especially necessary when the target language is morphologically rich, like German, whose words are usually compounded with sub-words expressing different meanings and grammatical structures" This claim should be verified, e.g. by comparing to English-French as well as model inspection.
> > > >
> > > >
> > > > The statistical analysis of the attention maps show a good percentage for token-to-phrase, phrase-to-token, and phrase-to-phrase in most of the layers of the model. We plan to compare with En - Fr in the the future.
> > > >
> > > >
> > > > We hope our responses are detailed and informative enough so that the reviewer can reconsider his judgements about our work. Thank you again for your review.

---

> > > > > ### Comment · AnonReviewer2 · 2018-11-21
> > > > > **Still not convinced**
> > > > >
> > > > > Thank you for the very detailed response!
> > > > >
> > > > > My main concern is still that there are multiple moving parts whose contribution is not clearly disentangled. Most strikingly, in Table 1, three different configurations obtain the best results on four datasets. Of course, there is not one method that will work best on very problem, but these are all translation tasks and for results to be convincing there should at least be a strong pattern.
> > > > >
> > > > > To summarize, I am still not convinced that the approach is really an improvement over the existing models and so keep with my original assessment.

---

> ### Author Response · Authors · 2018-11-20
> **Request for further reviews and comments and discussions**
>
> Dear reviewer,
> Thanks again for your initial comments.
> We have responsed to your concerns and revised/improved our paper accordingly.
> We hope you could spend some time to discuss more about the paper.
> We are eager to hear more advice, ideas and comments from you and have a discussion with you.

---

### Official Review · AnonReviewer1 · 2018-11-02
**This work starts from the premise that current models attends to unigram representation, which is wrong (keys and values already depends on multiple source/target positions). The empirical results are missing recent improvements. The reported empirical advantage compared to baseline is thin.**

**Rating:** 5
**Confidence:** 5

**Review:**

Phrase-Based Attention

Paper Summary:

Neural translation attention computes latent alignments which pairs input/target positions. Phrase-based systems used to align pairs of spans (n-grams) rather than individual positions, this work explores neural architectures to align spans. It does so by compositing attention and convolution operations. It reports empirical results that compares n-gram to uni-gram attention.

Review:

This paper reads well. It provides appropriate context. The equations are correct. It lacks a few references I mentioned below. The main weaknesses of the work lies in its motivation and in the
empirical results.

This work motivation ignores an important aspect of neural MT: the vectors that attention compares (“queries” and “keys”) do not summarizes a single token/unigram. These vectors aggregate information across nearby positions (convolutional tokens, Ghering et al 2017), all previous tokens (recurrent models, Suskever et al 2014) or the whole source sentence (transformer, Vaswani et al 2017). Moreover multiple layers of attention are composed in modern decoders, comparing vectors which integrates information from both source and target. These vectors cannot be considered as the representation of a single unigram from the source or from the target.

The key-value convolution method 3.1.1 is not different from (Ghering et al 2017) which alternates computing convolution and attention multiple times. The query as kernel is a contribution of this work, it is highly related to the concurrent submission to ICLR on dynamic convolution “Pay Less Attention with Lightweight and Dynamic Convolutions”. This other work however reports better empirical results over the same benchmark.

On empirical results, it seems that the table does not include recent results from work on weighted transformer (Ahmed et al 2017) or relative attention (Shaw et al 2018). Also a 0.1 BLEU improvement over Vaswani et al seems brittle, is your result averaged over multiple runs, could the base transformer be better with as many parameters/updates as your model?

Review Summary:

This work starts from the premise that current models attends to unigram representation, which is wrong (keys and values already depends on multiple source/target positions). The empirical results are missing recent improvements. The reported empirical advantage compared to baseline is thin. The most interesting contribution is the query as kernel approach: however the concurrent submission “Pay Less Attention with Lightweight and Dynamic Convolutions” obtains better empirical results with a similar idea.

Missing references:

Karim Ahmed, Nitish Shirish Keskar, and Richard Socher. 2017. Weighted transformer network for machine translation. arxiv, 1711.02132.

Peter Shaw, Jakob Uszkoreit, and Ashish Vaswani. 2018. Self-attention with relative position representations. In Proc. of NAACL.

---

> ### Author Response · Authors · 2018-11-13
> **Response to reviewer's comment**
>
> Thank you for your insightful reviews. We address your comments as follows.
> 1/ This work motivation ignores an important aspect of neural MT: the vectors that attention compares (“queries” and “keys”) do not summarizes a single token/unigram. These vectors aggregate information across nearby positions (convolutional tokens, Ghering et al 2017), all previous tokens (recurrent models, Suskever et al 2014) or the whole source sentence (transformer, Vaswani et al 2017). Moreover multiple layers of attention are composed in modern decoders, comparing vectors which integrates information from both source and target. These vectors cannot be considered as the representation of a single unigram from the source or from the target.
>
> We are sorry if our writing gave you the wrong impression that the vectors for the tokens are computed independently. However, we did NEVER state that in our paper, rather we wrote (in the Introduction):
>
> “Despite the advantages, the concept of phrasal attentions has largely been neglected in NMT, as most NMT models generate translations token-by-token autoregressively, and use the token-based attention method which is order invariant. Therefore, the intuition of phrase-based translation is vague in existing NMT systems that solely depend on the underlying neural architectures (recurrent, convolutional, or self-attention) to incorporate compositional information.”
>
> To elaborate on this, what we mean by “token-based attention” is described by the formula: Softmax(QK^T)V, which is, by definition, order-invariant and token-based. This means that if we change the order of the vectors in Q, K and V, there is no difference in the resulting attention scores and the context vectors. However, we did not say that the inputs to this formula (Q, K, V) are order invariant (see Sec. 2). Indeed, they can be order-variant, inter-dependent and causal (in case of the decoder). Different architectures have their own ways to ensure that theses inputs have such characteristics before passing to the attention. For instance, recurrent cells encode all previous tokens to the current one, convolutional layers encodes nearby tokens to the current one, self-attention encodes all (or previous in case of decoder) vectors from Keys and Values embedded by positional encoding. We hope this clarifies the confusion, and now let us explain why phrasal attention is needed for neural machine translation.
>
> It is true that the underlying architectures encode vector representations for tokens by aggregating information across multiple locations. But these vectors represent the respective input “tokens” considering their context. This is similar to ELMo/BERT representation of the tokens where the nearby vectors provide context-relevant clues only to represent the current token, and one does not use this as a representation of the phrase, rather uses it as a representation of the corresponding token.
>
> Note that we do not use ELMo/BERT representation of the tokens directly for the prediction tasks, rather they are incorporated into a model to consider the task-specific dependencies often modeled as inductive biases. For example, for NER task, these representations are fed into a bi-LSTM-CRF to model dependencies not only the in the input representations but also in the output sequence (between NER tags). For SQuAD QA task, the representations are used in a bidirectional attention flow network (BiDAF, 2017) to model task-relevant structures. Similarly, for textual entailment, the vectors are used in the ESIM model (Chen et al. 2017) that uses a bi-LSTM encoder, followed by a matrix attention layer and inference layers. The main point here is that the token representations learned by ELMo/BERT are incorporated into a model for the target task. Our phrase-based attention methods provide a model for the NMT task to incorporate phrasal alignments (as explicit inductive bias in the NMT model), and this is orthogonal to the underlying representation learning neural architectures.
>
> Response continues to next comment.

---

> > ### Author Response · Authors · 2018-11-13
> > **Continued response**
> >
> > 2/ The key-value convolution method 3.1.1 is not different from (Ghering et al 2017) which alternates computing convolution and attention multiple times. The query as kernel is a contribution of this work, it is highly related to the concurrent submission to ICLR on dynamic convolution “Pay Less Attention with Lightweight and Dynamic Convolutions”. This other work however reports better empirical results over the same benchmark.
> >
> > Good observation! The key-value convolution method is similar to (Ghering et al 2017) when it is considered only as a stand-alone attention method, as used for (single-head) cross attention in their work. In our phrasal attention, the key-value convolution is used for both self- and cross-attentions, and more importantly, it is used incorporation with homogeneous, heterogeneous or interleave heterogeneous multi-head attention, and  these are the techniques that make key-value convolution method effective for phrasal attention. We claim these techniques to be crucial contributions as well. We would like to emphasize that the two proposed phrase-based attention methods (ConvKV and QueryK) incorporated within the proposed architectures (homogeneous, heterogeneous, and interleaved) provide novel and complete solutions to phrasal attentions that as a whole is a significant contribution to the community.
> >
> > Thanks for the pointer to the “Pay Less Attention with Lightweight and Dynamic Convolutions” paper. We would like to emphasize the difference between their training settings and ours. As mentioned in our response to Reviewer 3, the Transformer has been well-known for its practical susceptibility to model size and batch size, and people have found that larger batch size generally yields higher performance; please see the discussion  https://github.com/tensorflow/tensor2tensor/issues/444 or this material https://ufal.mff.cuni.cz/pbml/110/art-popel-bojar.pdf for more details. We also observed similar phenomena in our experiments with both the transformer and the phrasal attention-based transformer.
> > Due to limited hardware resources (which is common in most academic labs), we could only afford to experiment with an 1-GPU setting, i.e., much smaller batch size. However, we used the same training settings in all of our experiments, including the baseline. For a fair comparison, we compare our model with the 1-GPU base transformer (26.07) using the same code provided by the authors (instead of the original 8-GPU base transformer (27.30), which we could not reproduce with a 1-GPU machine). Hence the gain margin we claim is 1.33 BLEU, not 0.1!
> >
> > In contrast, the “Pay Less Attention with Lightweight and Dynamic Convolutions” paper conducted their experiments with a Big model with hidden layer dimensions of 1024 in a massive system using 32 GPUs (16 times larger batch size than ours), while we could only run a model in a 1-GPU system with hidden layer dimensions of 512. Without speaking about the novelty, comparing ours with that paper is very unfair, and again we urge our reviewers not to penalize our work for not having industry-scale GPU facilities, rather evaluate it based on its scientific merits.
> >
> > Having said that, that paper is submitted at the same time as ours to the same conference. Both groups came up with different solutions to a problem, and got different results independently. Again, we would like to emphasize that Query-K is just one of our five main contributions. We believe that using their paper (or other papers submitted to the same conference at the same time) to disqualify ours is unfair. We believe conferences should value contribution, novelties and scientific merits that the papers offer, instead of comparing and contrasting with currently submitted papers, as happens in a competition.
> >
> > Response continued in next comment

---

> > > ### Author Response · Authors · 2018-11-13
> > > **Continued response**
> > >
> > > 3/ On empirical results, it seems that the table does not include recent results from work on weighted transformer (Ahmed et al 2017) or relative attention (Shaw et al 2018). Also a 0.1 BLEU improvement over Vaswani et al seems brittle, is your result averaged over multiple runs, could the base transformer be better with as many parameters/updates as your model?
> > >
> > > Thanks for the references. We were aware of the work on weighted transformer (Ahmed et al 2017), but since this work has not been accepted yet to any peer reviewed conference or journal (to the best of our knowledge), we did not include it in our paper. We have now included the suggested results in the revised version of the paper. In addition, we have also included new results on English-to-Russian and Russian-to-English translation tasks, where we observe even more improvements in BLEU scores compared to the transformer base (1.75 and 1.35, respectively).
> > >
> > > Summary of results on English-Russian
> > >
> > > For En-Ru:
> > > Base transformer:                                                         35.64
> > > Homogeneous QueryKernel:                                        36.31
> > > Heterogeneous ConvKV with n-gram (1-2):                 36.81
> > > Heterogenenous QueryKernel with n-gram (1-2-3-4):  37.39
> > > Interleaved heterogeneous QueryKernel (1-2):            37.24
> > >
> > > For Ru-En
> > > Base transformer:                                                         34.56
> > > Homogeneous ConKV:                                                 34.75
> > > Heterogenenous ConvKV with n-gram (1-2):               35.10
> > > Heterogenenous ConvKV with n-gram (1-2-3-4):         35.91
> > > Interleaved heterogeneous ConvKV (1-2):                   34.70
> > >
> > > We also conducted more experiments as per your request:
> > > transformer base  (6 layers):            63M params,        26.08 BLEU
> > > transformer big (6 layers):                214M params,      26.63 BLEU
> > > transformer base (10 layers):           91M params,        26.60 BLEU
> > > heterogeneous 4 layers (ConvKV):  60M params,        26.63 BLEU
> > > heterogeneous 6 layers (ConvKV):  80M params,        27.04 BLEU
> > >
> > >
> > > These results suggests that the improvements are not just random. However, we also ran our interleaved heterogeneous model on the EN-DE translation task for another seed. The results we got are:  27.33 (seed 100) and 27.32 BLEU (seed 20). We can see that the BLEU scores are quite consistent.
> > >
> > > As stated above, we compare our models with the transformer base model (26.07 BLEU), which is trained in identical settings as ours (1 GPU) for a fair comparison. Hence the gain margin we claim is 1.33 BLEU, not 0.1. Comparing our 1-GPU results with 8-GPU (and saying that it has only 0.1 gain) is unfair to us. Having said that, it would be nice if a third-party with sufficient resources could train our models at a massive scale (8-32 GPUs with larger batch size) to see how it performs in comparison with the state-of-the-art. We would be happy to share the codes.
> > >
> > > 4/ Reference: Karim Ahmed, Nitish Shirish Keskar, and Richard Socher. 2017. Weighted transformer network for machine translation. arxiv, 1711.02132.
> > >
> > > Peter Shaw, Jakob Uszkoreit, and Ashish Vaswani. 2018. Self-attention with relative position representations. In Proc. of NAACL.
> > >
> > > We have included these references in our revised version.
> > >
> > > We hope our responses are detailed and informative enough so that the reviewer can reconsider his judgements about our work. Thank you again for your review.

---

> ### Author Response · Authors · 2018-11-20
> **Request for further comments, reviews**
>
> Dear reviewer,
> Thanks again for your initial comments.
> We have responsed to your concerns and revised/improved our paper accordingly.
> We hope you could spend some time to discuss more about the paper.
> We are eager to hear more advice, ideas and comments from you and have a discussion with you.

---

### Official Review · AnonReviewer3 · 2018-11-06
**Cool Idea, More Evidence Needed**

**Rating:** 5
**Confidence:** 4

**Review:**

This paper presents an attention mechanism that computes a weighted sum over not only single tokens but ngrams (phrases). Experiments on WMT14 show a slight advantage over token-based attention.

The model is elegant and presented very clearly. I really liked the motivation too.

Having said that, I am not sold on the claim that phrasal attention actually helps, for two reasons:

1) The comparison to previous results is weak. There are more datasets, models, and hyperparameter settings that need to be tested.

2) Phrasal attention essentially adds an additional convolution layer, i.e. it adds parameters and complexity to the proposed model over the baseline. This needs to be controlled by, for example, adding another transformer block to the baseline model. The question that such an experiment would answer is "does phrasal attention help more than an extra transformer layer?" In my view, it is a more interesting question than "does phrasal attention help more than nothing?"

Also related to concern (2), I think that the authors should check whether the relative improvement from phrasal attention grows/shrinks as a function of the encoder's depth. It could be that deep enough encoders (e.g. 10 layers) already contain some latent representation of phrases, and that this approach mainly benefits shallower architectures (e.g. 2 layers).

===  MINOR POINTS ===
If I understand the math behind 3.1.2 correctly, you're first applying a 1x1 conv to K, and then an nx1 conv. Since there's no non-linearity in the middle, isn't this equivalent to the first method? The only difference seems to be that you're assuming the low-rank decomposition fo the bilinear term at a different point (and thus get a different number of parameters, unless d_k = d_q).

Have you tried dividing by sqrt(d_k * n) in 3.1.1 too?

While the overall model is well explained, I found 3.3 harder to parse.

---

> ### Author Response · Authors · 2018-11-13
> **Response to reviewer's comment**
>
> Thank you for your insightful reviews. We address your comments as follows.
>
> 1/ The comparison to previous results is weak. There are more datasets, models, and hyperparameter settings that need to be tested:
>
> We agree that more experiments with other datasets, model variants and hyperparameter settings would make the paper stronger. Due to the lack of GPU machines (which is common in an academic setting), we could perform only a handful of experiments before the submission deadline. However, we continued performing the intended experiments after the deadline. In particular, we have experimented with the English-to-Russian and Russian-to-English translation tasks (i.e., more datasets), and explored other model variants and hyperparameters (e.g., higher order ngrams). We revised our paper accordingly. Here is the summary of the new results for your consideration:
>
> For En-Ru:
> Base transformer:                                                              35.64
> Homogeneous QueryKernel:                                           36.31
> Heterogeneous ConvKV with n-gram (1-2):                  36.81
> Heterogenenous QueryKernel with n-gram (1-2-3-4):37.39
> Interleaved heterogeneous QueryKernel (1-2):           37.24
>
> For Ru-En
> Base transformer:                                                             34.56
> Homogeneous ConKV:                                                     34.75
> Heterogenenous ConvKV with n-gram (1-2):               35.10
> Heterogenenous ConvKV with n-gram (1-2-3-4):         35.91
> Interleaved heterogeneous ConvKV (1-2):                     34.70
>
> We also conducted more experiments as per your request:
> transformer base  (6 layers):            63M params,        26.08 BLEU
> transformer big (6 layers):                214M params,      26.63 BLEU
> transformer base (10 layers):           91M params,        26.60 BLEU
> heterogeneous 4 layers (ConvKV):  60M params,        26.63 BLEU
> heterogeneous 6 layers (ConvKV):  80M params,        27.04 BLEU
>
> As you may see, transformer big and transformer base (10-layers) have more parameters; they should perform better than transformer base. But the margins are not significant because of limited batch size (explained later). On the other hand, our heterogeneous model with just 4 layers is already on par with transformer base 10 layers or transformer big, though it has less parameters.
>
> Please note that the original transformer paper (Vaswani et al. 2017) conducted their experiments at a more massive scale (base and big models were trained with 8 GPUs) than what we could afford in a common academic lab. There have been many evidences that practical training of the transformer networks (theirs and ours) is significantly susceptible to the batch size (which increases with the number of GPUs used), and training a 1-GPU setup for sufficiently long does not produce similar results as an 8-GPU setup; please see the discussion  https://github.com/tensorflow/tensor2tensor/issues/444 or this material https://ufal.mff.cuni.cz/pbml/110/art-popel-bojar.pdf for more details.
>
> We compare our model with a version of the transformer that is trained on a single GPU using the same setting and code provided by the authors. In order to make a fair comparison to the baseline, we kept all of our experimental settings identical across all experiments. Thus, the comparison of our model (27.40 BLEU) should be made with this (1-GPU) baseline (26.07 BLEU) as opposed to the reported results (27.30 BLEU with 8 GPUs) in the original paper, because we believe that is not a fair comparison.
>
> With due respect, to comment on your statement “Experiments on WMT14 show a slight advantage over token-based attention”, 1.3 BLEU improvements in English-to-German translation task is quite large (as acknowledged by Reviewer 2). Our new experiments on English-Russian translation tasks show even larger gains (up to 1.75 BLEU) compared to the baseline. We urge our reviewers not to penalize our work for not having industry-scale GPU facilities, rather evaluate it based on its scientific merits. We think, the two proposed phrase-based attention methods (ConvKV and QueryK) incorporated within the proposed architectures (homogeneous, heterogeneous, and interleaved) provide novel and complete solutions to phrasal attentions that as a whole is a significant contribution to the community.
>
> Response continue in the next comment.

---

> > ### Author Response · Authors · 2018-11-13
> > **Continued response**
> >
> > 2/ Phrasal attention essentially adds an additional convolution layer, i.e. it adds parameters and complexity to the proposed model over the baseline: does phrasal attention help more than an extra transformer layer. Also related to concern (2), I think that the authors should check whether the relative improvement from phrasal attention grows/shrinks as a function of the encoder's depth. It could be that deep enough encoders (e.g. 10 layers) already contain some latent representation of phrases, and that this approach mainly benefits shallower architectures (e.g. 2 layers).
> > Good question! It is true that phrasal attentions introduce extra layers, but they are designed differently with a different role to play in the model. In particular, phrasal attentions are designed to explicitly model ngram relations, while adding more layers to the transformer only increases model capacity and does not necessarily model ngram relations explicitly.
> >
> > We argue that the extra layers added to the original transformer are not explicitly instructed to exploit phrasal relations, though they are theoretically capable of. One way an attention layer can exploit phrases is by assigning comparably high softmax scores to the n-grams of the phrase. However, there is hardly any evidence that it really does so, in contrast, it usually concentrates the scores on unigrams. Please have a look at the attention heatmap provided in (Vaswani et al., 2017). On the contrary, the attention heatmaps and the statistical analysis of phrasal attentions provided in our paper show that the attention scores choose to concentrate on the phrase-level layers instead of the original transformer (unigram) layer, even when we place them side by side within a global Softmax (heterogeneous).
> >
> > We applied phrasal attentions with the same number of layers as the original transformer (6 layers), which we believe is not shallow. Adding more layers also increases linearly the time complexity of the model, while we aim to maintain the same parallelizability of the original approach.
> >
> > Having said that, we ran an experiment with the original transformer to test if adding more layers (total of 10 layers in the encoder) does indeed result in better performance. The results are reported above.
> >
> > 3/ [Minor] if convkv and query-kernel is the same
> >
> > There are two key differences: (1) the order by which the element-wise dense (1x1 convolution) and the convolution operations are applied, and (2) more importantly, the dynamics of the query.
> >
> > First method (ConvKV): we apply nx1 convolution with weight $W_k$ (nx1) to K, and 1x1 convolution (dense) on Q  with weight $W_q$. In this case, the query interacts with a summarized version (a vector) of the associated key vectors representing a phrase.
> >
> > Second method (QueryK): we apply 1x1 convolution with weight $W_k$ (1x1) to K, and then nx1 convolution with query as the kernel weight. In this case, the query interfaces directly with n vectors representing a phrase (receptive field). Since the query is changing, the filter applied to the receptive field is dynamic (as opposed to a fixed weight).
> >
> > In fact, the core idea of the second method is similar to another ICLR-19 submitted paper (as pointed out by Reviewer1): Pay Less Attention with Lightweight and Dynamic Convolutions (https://openreview.net/forum?id=SkVhlh09tX). The key idea here is to use dynamic kernel for convolution.
> >
> > 4/ Have you tried dividing by sqrt(d_k * n) in 3.1.1 too?
> > We tried and they have no or minor differences.
> >
> > We hope our responses are detailed and informative enough so that the reviewer can reconsider his judgements about our work. Thank you again for your review.

---

> ### Author Response · Authors · 2018-11-20
> **Request for further questions, comments.**
>
> Dear reviewer,
> Thanks again for your initial comments.
> We have responsed to your concerns and revised/improved our paper accordingly.
> We hope you could spend some time to discuss more about the paper.
> We are eager to hear more advice, ideas and comments from you and have a discussion with you.

---

### Meta-Review · Area_Chair1 · 2018-12-02
**Reject**

**Confidence:** 4
**Recommendation:** Reject

**Metareview:**

All reviewers agree in their assessment that this paper does not meet the bar for ICLR. The area chair commends the authors for their detailed responses.